# Evolutionary Dynamics of Foot and Mouth Disease Virus Serotype A and Its Endemic Sub-Lineage A/ASIA/Iran-05/SIS-13 in Pakistan

**DOI:** 10.3390/v14081634

**Published:** 2022-07-26

**Authors:** Syeda Sumera Naqvi, Nazish Bostan, Katsuhiko Fukai, Qurban Ali, Kazuki Morioka, Tatsuya Nishi, Muhammad Abubakar, Zaheer Ahmed, Sadia Sattar, Sundus Javed, Aamira Tariq, Asma Sadiq

**Affiliations:** 1Molecular Virology Laboratory, Department of Biosciences, Comsats University Islamabad, Islamabad 45550, Pakistan; s.sumeranaqvi@gmail.com (S.S.N.); sadia.sattar@comsats.edu.pk (S.S.); 2Exotic Disease Research Station, National Institute of Animal Health, National Agriculture and Food Research Organization, 6-20-1 Josui-honcho, Kodaira 187-0022, Japan; fukai@affrc.go.jp (K.F.); morioka@affrc.go.jp (K.M.); ultra1124@affrc.go.jp (T.N.); 3National Veterinary Laboratory, Ministry of National Food Security & Research, Islamabad 45710, Pakistan; drqurban@yahoo.com (Q.A.); mabnvl@gmail.com (M.A.); 4World Health Organization, Islamabad 45500, Pakistan; 5Reagent and Vaccine Services Section, Foreign Animal Disease Diagnostic Laboratory, Animal and Plant Health Inspection Service, National Veterinary Services Laboratories, United States Department of Agriculture, Plum Island Animal Disease Centre, Orient Point, Orient Point, Southold, NY 11957, USA; zaheer.ahmed@usda.gov; 6Microbiology and Immunology Laboratory, Department of Biosciences, Comsats University Islamabad, Islamabad 45550, Pakistan; sundus.javed@comsats.edu.pk (S.J.); aamira_tariq@comsats.edu.pk (A.T.); 7Department of Microbiology, University of Jhang, Jhang 35200, Pakistan; asmasadiq@uoj.edu.pk

**Keywords:** adoptive evolution, foot and mouth disease virus, next-generation sequencing, phylogeography, Pakistan, recombination, serotype A

## Abstract

Foot and mouth disease (FMD) causes severe economic losses to the livestock industry of endemic countries, including Pakistan. Pakistan is part of the endemic pool 3 for foot and mouth disease viruses (FMDV), characterized by co-circulating O, A, and Asia 1 serotypes, as designated by the world reference laboratory for FMD (WRL-FMD). FMDV serotype A lineage ASIA/Iran-05 is widespread in buffalos and cattle populations and was first reported in Pakistan in 2006. This lineage has a high turnover, with as many as 10 sub-lineages reported from Pakistan over the years. In this study, we reconstructed the evolutionary, demographic, and spatial history of serotype A and one of its sub-lineages, A/ASIA/Iran-05/SIS-13, prevalent in Pakistan. We sequenced nearly complete genomes of three isolates belonging to sub-lineage A/ASIA/Iran-05/SIS-13. We estimated recombination patterns and natural selection acting on the serotype A genomes. Source and transmission routes in Pakistan were inferred, and the clustering pattern of isolates of the SIS-13 sub-lineage were mapped on a tree. We hereby report nearly complete genome sequences of isolates belonging to sub-lineage A/ASIA/Iran-05/SIS-13, along with purported recombinant genomes, and highlight that complete coding sequences can better elucidate the endemic history and evolutionary pressures acting on long-term co-circulating FMDV strains.

## 1. Introduction

Foot and mouth disease virus (FMDV) is a highly contagious positive-sense single-stranded RNA virus that belongs to the family *Picornaviridae*, and it is associated with an acute vesicular disease in cloven-hoofed livestock. FMDV has a diverse host range, affecting both domesticated and wild animal species, and has established endemic and transboundary transmissions in many countries of the Middle East, Africa, and Asia [1]. Seven FMDV pools are identified by the World Reference Laboratory for FMD (WRLFMD), United Kingdom (UK). Pakistan has the endemic pool 3 of FMDV, which is characterized by the presence of three serotypes: O, A, and Asia 1, co-circulating among cloven-hoofed animal populations [2]. Co-infection by these serotypes has also been reported in various studies [3,4,5,6].

FMDV possesses 8.5 kb genome, having 5′ and 3′ untranslated regions (UTR), along with a single open reading frame (ORF) containing L coding region for leader proteinase P1 region coding for four structural (capsid) proteins (VP1–VP4), along with P2, and P3 regions coding for non-structural proteins [7]. The virus is conventionally characterized into serotypes and lineages based on multiple signature amino acid substitutions and deletions in the highly variable 1D (VP1) region of capsid protein, which is an important target for host receptor binding and immune response [8]. The P1 region of different serotypes possesses four to five antigenic sites on VP1, VP2, and VP3 [9,10].

FMDV has high mutation rates, possesses a compact genome structure, and is dependent on the host cell machinery. Its genome, like other RNA viruses, undergoes continuous evolutionary changes through the processes of mutation and recombination. RNA viruses usually derive their evolution through the strong purifying and sporadic positive evolution to ensure their survival during successive reproductive cycles [11,12]. These measures, along with differential host preference of FMDV topotypes, influence their persistence and prevalence in susceptible host populations [13,14]. The virulence and transmissibility of co-circulating viral strains in diverse and susceptible hosts lead to flux in the virus population over time [15]. This leads to the emergence and establishment of new variants that can compromise vaccine efficacy and have direct implications for disease control through preventive vaccination [16,17].

Pakistan has an agriculture-based economy, with 8 million families directly or indirectly associated with the livestock industry, which contributes about 11.7% of the country’s gross domestic product (GDP) annually [18]. The cattle (~50 millions) and buffalo (~40 millions) populations are generally at risk of getting FMD infection, as high morbidity is reported in these animals [19]. In Pakistan, the FMDV outbreaks are contained by mass vaccination with polyvalent vaccines containing antigens of A, O, and Asia 1 serotypes, along with quarantine measures, as culling is not considered a viable option. The persistence of FMDV negatively affects the livestock industry due to loss in meat and milk production and global trade restrictions on the export of dairy products from endemic areas. Rapid population growth has resulted in increased demand for livestock products. This demand is accommodated by the establishment of extensive urban farming systems, in addition to the traditional methods. A major challenge for such units is sustainable and efficient farm management. A failure of management can lead to the emergence and subsequent spread of new and possibly more infectious variants of FMDV in an area [17,20]. Serotype A is included in all seven endemic pools of FMD, except pool 6, and is differentiated into three topotypes: ASIA, AFRICA, and EURO-SA. A diverse and persistent strain of topotype ASIA is lineage Iran-05, prevalent in the Middle East and South Asian countries, with 16 reported sub-lineages [21,22]. This strain is also prevalent in Pakistan and was first reported in 2006 as topotype ASIA and lineage Iran-05 [23]. Over the years, as many as ten sub-lineages (characterized and differentiated by >5% nucleotide difference in VP1) of Iran-05 have been reported from Pakistan that co-circulated and dominated for a certain period and then either disappeared or were replaced by the other sub-lineages: SIN-08 (2008), ESF-10 (2008–2009), Bar-08 (2009–2010), FAR-09 (2009), AFG-07 (2008–2011), SIS-10 (2010), SIS-12 (2012–2013), HER-10 (2011–2012), FAR-11 (2011–present), SIS-13 (2014–present). Currently, different sub-lineages of Iran-05 are circulating in Pakistan, as reported in the WRL-FMD country reports. Two of these sub-lineages are designated as SIS-13 and FAR-11; others are not yet designated [24].

FMDV serotype A, sub-lineage A/ASIA/Iran-05/SIS-13, has established its presence in Iran, Pakistan, and Afghanistan. This region provides a unique environment for the emergence and establishment of new variants of serotypes O, A, and Asia 1 [22]. Although an extended geographical context is more desirable and informative, it is equally important to keep track of ongoing changes in a niche that are presumed to contribute to the global diversity of FMDV. This study was designed to fill in the gaps and document the findings in the more restricted geographical zones of Pakistan. We report nearly complete genome sequences of isolates of FMDV serotype A sub-lineage A/ASIA/Iran-05/SIS-13. The origin, evolutionary pressures, and phylodynamics are investigated in comparison to publicly reported FMDV variants given in the National Center for Biotechnology Information (NCBI) GenBank. Overall, our results reinforce the knowledge of continuously evolving variants that persist endemically and pose a significant threat that needs to be readdressed with the perspective of changing climate and global indicators.

## 2. Materials and Methods

### 2.1. Virus Isolation

Epithelium samples, collected from infected cattle suspected of FMD through clinical diagnosis, are sent to the National Veterinary Laboratory (NVL), Islamabad, for confirmation of the causative agent under the national surveillance program for the control and prevention of FMDV. The detection of FMDV and its serotypes is routinely performed through sandwich ELISA (IZLER, Brescia, Italy; Pirbright institute, UK) using 10% epithelium suspension in Phosphate Buffer Saline (PBS). For virus isolation, the samples were prepared in Dulbecco MEM supplemented with 10% Fetal bovine serum and 1% L-Glutamine. The 10% suspensions were prepared in Biomashar II closed system disposable tissue homogenizers and clarified at 3000 rpm for 5 min and stored in 1.5 mL tubes. Freshly prepared supernatants (150 µL) were inoculated to 70–80% confluent LFBK-αvβ6 (USDA, ARS, PIDAC USA) [25] cell monolayer in T-75 tissue culture flasks and incubated at 37 °C and 5% CO_2_ for one hour; the flasks were agitated gently every 15 min. The flasks were washed twice and supplemented with maintenance media. After complete cytopathic effects (CPE) appearance, cell culture fluids were collected and clarified twice at 3000 rpm for 15 min and stored at −80 °C. The CPE positive isolates A_PK_C6_2017 and A_PK_C8_2018 were used for sequencing.

### 2.2. Genome Sequencing

Viral RNA was extracted from CPE positive cell culture supernatants using a High pure viral RNA extraction kit (Roche Diagnostics, Tokyo, Japan) and reverse-transcribed using Invitrogen™ SuperScript™ IV Reverse Transcriptase for the synthesis of cDNA (ThermoFisher Scientific, Waltham, MA, USA) with primers 2B331R (GGCACGTGAAAGAGACTGGAGAG) and 3′NT-Rc (CGCCTCAGAGTCTTTCTGCCAATTG), as per manufacturer’s protocols. The strand of cDNA was amplified using two primers sets 5′NT-Fa (CCGTCGTTCCCGACGTTAAAGGG) and 2B331R, 2B217F (ATGGCCGCTGTAGCAGCACGGTC), and 3′NT-Rc with PrimeSTAR^®^ Max DNA Polymerase (Takara, Shiga, Japan) to generate two PCR products of approximately 4 kb, which were purified using a QIAquick PCR purification kit (Qiagen, Germantown, MD, USA), and their concentration was measured using an Invitrogen^TM^ Qubit^TM^ dsDNA BR Assay kit (ThermoFisher Scientific, Waltham, MA, USA). For sequencing of these PCR products, a library was constructed by shearing PCR products into fragments using Ion Shear™ Plus Reagents Kit (ThermoFisher Scientific), which were then purified using Agencourt AMPure XP Reagent (Beckman Coulter, Pasadena, CA, USA). The fragments were nick repaired with Ion Plus Fragment Library kit (ThermoFisher Scientific) and ligated with a unique barcode for each sample using Ion Xpress Barcode adaptors 1-16 kit (ThermoFisher Scientific). The purified barcoded library was size-selected for size 350 bp with Invitrogen^®^ E-Gel SizeSelect II 2% Agarose gel (ThermoFisher Scientific cat #G661012). After purification, the barcoded library was amplified using Platinum PCR Supermix High fidelity with 8 PCR cycles. After amplification, the library was purified again and quantified using Agilent^®^ High Sensitivity DNA reagents on the Agilent 2100 Bioanalyzer system (Agilent Technologies, Santa Clara, CA, USA). Finally, a 26 pM library was prepared and loaded to the next-generation sequencing PGM platform Ion OneTouch™2 System (ThermoFisher Scientific). The reads were assembled and mapped to the reference sequences to generate consensus sequences within built-in Torrent suite software version 5 using default settings [26,27]. Nearly complete genomes (nucleotide number, 7744) with partial 5′ and 3′ regions, were determined for isolates A_PK_C6_2017 and A_PK_C8_2018 in this study. Two different consensus sequences were generated for A_PK_C8_2018. After confirmation of the presence of serotype A and serotype O by plaque purification and later VP1 sequencing, one clone for serotype O and one additional clone for serotype A were sequenced. The sequences for A_PK_C6_2017, A_PK_C8_2018, and A_PK_C8_2018_c1 were submitted to GenBank under accession numbers MZ493232, MZ493233, and MZ493234, respectively.

### 2.3. Data Retrieval and Sequence Analysis

We determined the topotype, lineage, and sub-lineages of our sequences through phylogeny reconstruction and nucleotide homology, with reference sequences for FMDV serotype A provided by the WRLFMD at Pirbright institute, UK. We retrieved a sequence dataset consisting of 170 VP1 sequences, 11 P1 regions, and 25 complete genomes of Pakistani FMDV serotype A isolates, available from NCBI GenBank, and used these for sequence analysis in Mega X version 10.2.2 [28]. Complete genomes and P1 sequences were aligned by ClustalW in Maga X, and individual genes were demarcated within the single ORF using FMDV A24/Cruzeiro/Brazil/55, accession No. AJ251476.1, as a reference for FMDV serotype A genes. The phylogeny of VP1 sequences was reconstructed, along with reference sequences, using the maximum likelihood (ML) method with the TPM2u + F + I + G4 model selected by Model Finder [29] using 1000 ultrafast bootstraps [30] in IQTree v 1.6.12 [31]. The tree was imported and rendered in Figtree v 1.4.4 [32]. We also imported and rooted the unrooted, maximum likelihood tree of VP1 sequences in TempEst v 1.5.3 [33] to detect the temporal signal in sequence data, which is measured by the regression of root-to-tip divergence with sampling time (Appendix A). The detection of a good temporal signal is important for phylogeographic analysis.

### 2.4. Recombination Analysis

The effects of selection and recombination on FMDV genome variability were determined from the Datamonkey Adaptive Evolution Server [34] using HyPhy (hypothesis testing using phylogenies), which provides comparative methods and tools to estimate recombination and natural selection by fitting the codon-based evolutionary models on multiple sequence alignments (MSA) of protein-coding genes. For recombination analysis, complete coding sequences of A, O, and Asia 1 serotypes (Appendix A) were subjected to a genetic algorithm recombination detection (GARD) [35] test to screen for recombination events by searching for potential breakpoints. If a recombination signal was detected, the data was analyzed by Recombination Detection Program (RDP) v 4.1.01, which employs default programs RDP, GENECOV, Chimaera, MaxChi, Bootscan, and Siscan to infer recombination patterns within the genome. The acceptance was limited to recombination events detected by six methods (*p* < 0.001) after Bonferroni multiple testing correction. The final check was performed by subjecting the potential recombinants to similarity plotting and boot scanning in Simplot v3.5, along with serotype O (sub-lineage ANT-10) and Asia 1 (lineage Sindh-08) complete sequences.

### 2.5. Natural Selection Analysis

We used sequence-based neutrality tests to elucidate the effects of selection on the FMDV genomes in our dataset using DnaSP v6. To assess if any changes in the capsid proteins are the result of adaptive evolution, we used the two most widely used sets of tests. The first one analyzes the patterns and frequency of polymorphic sites in the sequences provided in DnaSP v6. We used Tajima’s D, Fu, and Li’s D and F methods that make inferences about selection based on low, intermediate, and high-frequency polymorphism in a population. The VP1 gene sequences were subjected to coalescent simulations (1-locus|1-population model) using the standard neutral model (SNM) with 10,000 replicates. The other set draws the results from non-synonymous to synonymous ratios at polymorphic sites. We used the methods BUSTED, aBSREL, MEME, FEL and FUBAR to test if any gene, branch, or amino acid site is under sporadic or pervasive positive and negative selection. BUSTED [36] and aBSREL [37] tests for natural selection acting on a gene and the branch level, respectively. A mixed-effects random model of evolution (MEME) allows the site-wise episodic diversification estimation by allowing the variation in selective pressure rates at each site. The fixed effects likelihood (FEL) method was used to estimate the selection pressures on VP2 and VP3 genes. FEL is suitable for pervasive diversification and conservation detection in the small-size datasets (in our case, 20 unique sequences). It applies bootstrap resampling and likelihood confidence intervals for each variable site. Fast unconstrained Bayesian approximation (FUBAR) is also a powerful tool in the suite of Hyphy that analyses pervasive diversifying selection in a population by Bayesian inference. These models accommodate external to internal branch ratios to avoid dN/dS ratio bias on population evolution inference due to transient changes in external branches. The branch lengths in the phylogeny are optimized, and hypothesis support is tested using the likelihood ratio test [12,38,39,40,41]. The significant values were set to *p* < 0.05 and for FUBAR, posterior probabilities were set to >0.98.

### 2.6. Phylogeographic Analysis

After detection of a good temporal signal in TemPest, we used spatial and time-stamped sequences to generate a time-calibrated maximum clade credibility (MCC) phylogenetic tree by Bayesian inference usung Beast v 1.10.4 [42]. This software allows the simultaneous inference of evolutionary, demographic, and phylogeographic history. This requires prior information about partition scheme and substitution, molecular clock, and tree model. The partition scheme 2 (1 + 2, 3) and GTR + Γ + I substitution model were selected in Partition Finder 2.1.1 [43] to fit the data. The molecular clock was selected using Path sampling/Generalized steppingstone (GSS) in Beast v 2.6.6 by log marginal likelihood estimation (MLE) of three models: strict clock, uncorrelated relaxed clock lognormal (UCLN), and uncorrelated relaxed clock exponential. The difference in log MLE (>150) strongly supported the UCLN molecular clock choice that accommodates rate heterogeneity across tree branches and lineages [44]. We used a nonparametric coalescent Bayesian skygrid demographic model with a cut-off setting of k > root height (we set k as 15). This model allows the estimation of changes in effective population size N_e_ over time at specific grid points [45]. The asymmetric discrete phylogeographic model was used, with a separate partition for discrete traits (locations: provinces) using a continuous-time Markov chain (CTMC) as the prior clock rate and enabling Bayesian stochastic search variable selection (BSSVS) to infer transmission between every pair of locations [46]. The two independent chains of Markov chain Monte Carlo (MCMC) were run for 200 million iterations, and data logs were combined in Log Combiner v 1.10.4. The combined trace file was visualized in Tracer v 1.7.1 [47] to assess the satisfactory sampling and mixing of posteriors (effective sample size > 200). The MCC tree was generated from trees logged with 10% burn-in in Tree Annotator v 1.10.4 [42], and the tree was visualized in Figtree v 1.4.4. The transitions and diffusion routes between locations were summarized in SpreaD3 [48] and rendered in Google Earth Pro (©2021 LLC).

## 3. Results

### 3.1. Sequence Analysis

We reconstructed the phylogeny of 206 VP1 sequences of serotype A isolates from Pakistan with reference sequences of 11 sub-lineages of Iran-05. The sequences which clustered with a reference genome were assigned a sub-lineage (Figure 1). The nucleotide identity of all isolates with reference sequences was also confirmed through the NCBI Blastn. Our isolates, along with 30 taxa from GenBank, clustered with A/ASIA/Iran-05/SIS-13, a reference genome A/IRN/27/2013 [22]. The isolates belonging to sub-lineage AFG-07 were split into two clades that each seemed to give rise to a different evolutionary pathway, although both sets of isolates have close nucleotide similarities to reference sequence FJ755007 isolated in 2007 from Afghanistan [49]. All other isolates were clustered with their respective reference genome, with clade support of more than 90% ML bootstrap. SIS-13 isolates collected in 2018 formed a sister clade, and a comparison of their VP1 sequences with VP1 of other SIS 13 isolates revealed changes at amino acid sites H/R28Q, E95V, K109R, P149S, Q168R, and Q198S, while five different amino acid substitutions were found across VP1 of 2018 isolates at sites 85, 171, 176, 182, 197.

The sequences of three FMDV isolates from this study were further analyzed for putative non-synonymous nucleotide substitutions and amino acid changes in capsid protein region P1 (VP1–VP4) along with P1 regions of previously isolated Pakistani isolates, twenty-five complete genomes listed in Appendix A and 11 P1 sequences for serotype A isolates, that belong to 8 sub-lineages of A/ASIA/Iran-05, are shown. The majority of the isolates from sub-lineages HER-10, ESF-10, BAR-09, and SIN-08 were spatially and temporally related to each other, with a difference of a few nucleotide substitutions, and appear to belong to similar outbreaks. The changes in variable sites of the P1 region are listed in Table 1 and Table 2; however, VP4 is not represented, as in this region, only two singleton amino acid substitutions were observed in two different isolates. There were no changes in amino acid substitution in four antigenic sites of the capsid region (previously reported for serotype A), except at VP2 residue 72, where a synonymous substitution was found (Figure 2). The comparison of A_PK_C6_2017 (MZ493232) and A_PK_C8_2018 (MZ493233) with each other identified 46 sites in the polyprotein with genome-wide amino acid substitutions, except for VP4 and 2A, where no amino acids substitutions were observed. A schematic representation of amino acid changes in antigenic sites (Site I-IV) of P1 regions (VP1-VP3) common to the majority of the isolates is given in Figure 2.

### 3.2. Recombination Analysis

To analyze the extent of recombination and identify putative recombinant sequences in the dataset, complete genome sequences of FMDV isolates from three serotypes (A, O, and Asia 1) were subjected to GARD analysis. Details of the isolates are given in Appendix A. This algorithm identified 12 putative breakpoints and generated trees for each partitioned fragment. The most divergent parts of the genome are VP2, VP3, and VP1, as suggested by the clustering of individual fragments; however, no well-supported breakpoints were detected in these segments, except for the N-terminus of VP2. The clustering of individual genome fragments (genome coordinates: 3014–3464, 3465–4112, 4113–4895, 4896–5360, 5361–5924, 5925–6764, 6765–7008) was different from the serotype-specific clustering of the P1 region.

The recombination was further validated in RDP4 software, which utilized a wide range of methods to detect recombination and could identify breakpoints with global and local confidence intervals. This analysis identified a breakpoint hotspot pair (with 95% confidence interval) at about 900 and 2300 bp positions in the dataset (Appendix A). This analysis identified one isolate of our study (A_PK_C6_2017; Acc. No MZ493232) as a potential recombinant, along with three other isolates already reported from Pakistan (As/SIN/PAK/L2810/2009 [5], O/PAK/44/2008, and O/PAK/45/2008 [50]). The recombination signal for A_PK_C6_2017 was strong enough to be accepted with the support of all default programs (*p* < 0.001). Three fragments of A_PK_C6_2017 (3465–4112, 4113–4895, and 5361–5924) were clustered with O serotype isolates (O/PAK/14/2017 (MH784405) and O/PAK/4/2017 (MH784404)), rather than with serotype A sequences. This suggests that isolate A_PK_C6_2017 is a potential recombinant of O and A serotypes. The results showed that isolate A_PK_C6_2017 carried the marks of recombination in the P2 and P3 regions of the genome (Figure 3a,b). The recombination signal was also observed for A_PK_C8_2018 in one of the events identified by RDP4, but it was not confirmed in Simplot, Figure 3c,d.

### 3.3. Natural Selection Analysis

To detect selection pressure on the capsid coding genes of our dataset, we analyzed these genes using Tajima’s D, Fu, and Li’s D and F methods (Materials and Methods). The results of this analysis were non-significant (*p*-value > 0.2), which indicated that the capsid genes may be evolving neutrally, and evidence for directional or balancing evolution is not available in this dataset. Moreover, the results of aBRSL and BUSTED did not indicate any sporadic or pervasive selection acting at the gene or branch level. We then tested for pervasive and sporadic selection acting on three sites in the capsid protein which might not be observable in the gene-based tests. In VP1 protein, 9 sites were found to be under episodic positive selection including 43, 45, 60, 96, 149, 150, 168, 171, and 182. Among these sites, 60, 150, and 182 were episodic, with a low proportion of branches under selection, and they appeared in the external branches of a tree, while other sites were under episodic diversifying selection, and were found along the internal branches. FUBAR identified two sites, 96 and 168, under pervasive positive selection, and 77 under pervasive purifying selection pressure, along the internal branches, with a posterior probability of >0.98. The FEL test identified purifying, diversifying, neutral, and invariable sites in VP1 sequences along the phylogeny. The N-terminus and C-terminal of VP1 exhibit extreme residues under purifying selection. In the G-H loop of antigenic site I, all residues were invariable or evolving under neutral selection, except in the RGD motif (144, 145, 146), the D (146) was under pervasive purifying selection. The antigenic sites on VP1 were either invariable or evolving under neutral evolution, except site IV (148), which was under purifying selection (Figure 2). We ran FEL on 20 VP3 and VP2 sequences, and no sites were found under positive selection, while amino acids at position 14 and 17 in these genes in total 14 and 17 sites in these genes, respectively, were under negative selection.

### 3.4. Phylogeographic Analysis

The maximum likelihood tree, generated from spatially and temporally stamped serotype A isolates from Pakistan, had linearity in regression of root-to-tip divergence with sampling years (R^2^ = 0.79), indicating a strong temporal signal in the data (Appendix A). In serotype A, VP1 has accumulated genomic diversity at the mean evolutionary rate of 9.4 × 10³ (95% Bayesian credible interval (BCI) = 5.56 × 10³ − 0.014) substitution/site/year. The mean standard deviation of 0.03 (95% BCI = 0.009–0.01) indicated less variance across branches of the phylogenetic tree. The root of the tree trunk was dated to 2001.6 ± 6.7 (95% BCI = 1996.34–2005). The sub-lineage SIS-13 is evolving at an inferred rate of 3.2 × 10³ (95% BCI = 1.7 × 10³–5 × 10³); the substitution/site/year and time of the most recent common ancestor (tMRCA) for SIS-13 isolates was dated to year 2012.2 (95% BCI = 2011.1–2013.22). The ancestral location reconstruction with location annotated nodes of the MCC tree inferred Punjab province as a plausible ancestor location at the tree trunk, with posterior probability (pp) = 0.70 and for SIS-13 taxa, with pp = 0.96 (Figure 4).

The SIS-13 taxa were clustered with a posterior support of 1. The isolates from 2018 were clustered in a sister clade, with a posterior support of 1, and seem to originate from the same outbreak, with the oldest sequence isolated from Sargodha, a city in Punjab province. The reconstruction of the population trajectory plotted the effective population sizes (N_e_) 2005 onwards. In our analysis, we set the population parameter to predict at least three population changes per year. The N_e_ was increasing, with fluctuations in genetic diversity until 2015, then a steady decline was inferred until 2018, when the most recent samples were collected. The peaks are visible, hinting at cyclic trends of virus populations. The visible peak was highest around 2012, with a narrow 95% HPD, while the peak was lowest in 2005, with a wide 95% HPD. (Figure 5a). This disparity may be due to the difference in sampling intensity during this time. The viral lineage number was expanding logarithmically through time, until around 2012, when this trend become consistent (Figure 5b).

A parsimonious phylogeographic diffusion route was reconstructed from the MCC tree for seven discrete locations (provinces), Azad Jammu and Kashmir (AJ and K), Baluchistan, Gilgit Baltistan (GB), Islamabad Capital Territory (ICT), Khyber Pakhtunkhwa (KPK), Punjab, and Sindh. The province of Punjab was the most probable source, with a well-connected continuous transmission route between Punjab, Sindh, and ICT (95% credible set). Baluchistan, KPK, AJK, and GB provinces were sink locations, while Punjab, ICT, and Sindh provinces were inferred as source locations. The movement of lineages was mostly westward, with few transmissions between eastern regions, and lineages spent more time in Punjab (Figure 6). The diffusion rate of viruses among the provinces was 0.077/year. The locations’ rate matrix log generated with BSSVS was used to reconstruct transition rates between location pairs, which are inferred with the support of Bayes factors and posterior probabilities. No discernible trends in transition rates between locations were apparent (data not shown).

## 4. Discussion

We isolated and sequenced nearly complete genome sequences of foot and mouth disease viruses of sub-lineage SIS-13 that were collected from the province of Punjab in the years 2017 and 2018. We retrieved the relevant sequence data for serotype A from the NCBI GenBank database. The evolutionary analysis of the VP1 gene revealed that the FMDV serotype A population in Pakistan is evolving measurably at a rate of evolution of 9.2 × 10³ substitution/site/year that is comparable to the overall evolutionary rate of serotype A [15]. The SIS-13 sub-lineage is evolving at the rate of 3.2 × 10³, and the isolates from 2018 were more closely related to each other than to the viruses from 2014–2017 and have diverged from the reference sequence by about 6% in VP1. The 2018 isolates seem to originate from the same outbreak, with the oldest sequence isolated in Sargodha (Punjab), that later spread to other cities of Punjab. The SIS-13 strain of the Serotype A virus seemingly originated from the evolution of the AFG-07 sub-lineage (reference topotype sequence FJ755007) [49]. In our dataset, the AFG-07 isolates formed two clades, one with >95% boot strap repeat value and the another with 70% boot strap repeat value and seemed to follow separate evolutionary paths leading to different sub-lineages, including two current sub-lineages, i.e., FAR-11 and SIS-13 (Figure 1). The KT832824-37 isolates were previously reported as a new sub-lineage BAL-09 [50]; however, two closely related isolates of JX435107-8 (reported in the same study) were designated as AFG-07, according to the WRL-FMD classification system of FMDV [22]. All AFG-07 isolates have high nucleotide similarity with the reference sequence (>95%, which is required for the designation of a sub-lineage of FMDV).

The recombination analysis of complete coding sequences of A, O, and Asia 1 FMDVs revealed that the capsid (P1 region) of FMDVs is the most divergent region among its different serotypes, and it is free of inter-serotypes recombination. The P2 and P3 regions of A, O, and Asia 1 FMDVs have more sequence similarity with each other, and the presence of multiple breakpoints suggests the possibility that these regions are more likely to recombine as compared to the P1 region. The genome region that codes for 2C and 3C seemingly have the least serotype-specific clustering of all the fragments in this dataset. This is the reason that sequences of the P1 region are commonly used for the identification and designation of different serotypes of FMDVs. Four putative recombinant sequences were detected in the dataset; three were previously reported as putative recombinants [5,51]. The O/Pan-ASIA2/ANT-10 and ASIA 1/Sindh-08 viruses were reported to have putative recombinant genome regions acquired from A/ASIA/Iran-05. In these cases, the serotype A virus was the major parent contributing to the recombinant region. These recombinant sequences were discussed in detail in the study by Jamal et al. [51], so here we discuss sequences isolated in this study. One isolate in our study (A_PK_C6_2017) was inferred as a potential recombinant, with support from GARD and RDP4. The P1 and L^pro^ regions of A_PK_C6_2017 have serotype-specific clustering, but regions coding for 2A and 2B and 3A proteins were clustered with isolates O/PAK/14/2017 and O/PAK/4/2017, which were also isolated from the province of Punjab. A lack of experimental evidence makes the possibility of a cross-over event between the co-circulating strains inconclusive. Although we identified the co-infection of A_PK_C8_2018 with two serotypes A and O, we were not able to isolate any recombinant sequence. These results suggest that antigenically related viruses may carry slightly different non-structural proteins, but on the basis of our current results, we cannot suggest whether or not this will have any effect on disease manifestation.

In natural selection determining analyses, we did not find any polymorphism frequency-related patterns suggestive of directional or balancing evolution. BUSTED and aBSREL tests did not provide evidence of episodic diversification acting on the gene or branch of serotype A. However, many amino acid sites were observed under episodic diversification. A major section of the VP1 gene was evolving under purifying and neutral selection. The critical sites in the GH loop and C terminal of VP1 were mostly invariable in this dataset, and they have possibly evolved under neutral selection, except for strong purifying selection revealed at two sites, 146 and 148. In the capsid region, the amino acid residue 146 is a part of the RGD motif and is critical for host receptor binding. The amino acid residue 148 forms the neutralizing antigenic site IV, which is a highly conserved region that is probably formed by the interaction of the G-H loop with other capsid proteins [10]. In VP2 and VP3 proteins, many codons were evolving under purifying and conservative selection, and no evidence of episodic or pervasive diversifying selection along sites and branches was observed. In the polymorphic regions, many random substitutions were observed that were mostly evolving under neutral selection. These observations indicate that FMDV regulates genetic diversity to maximize survival by genetically changing as much as possible, while restricting disadvantageous mutations at critical sites. Sporadic diversification is a preferred method of evolution for many positive and negative single-stranded RNA viruses, which usually exhibit purifying selection acting on their genome due to their small sizes [12]. Genetic drift and spontaneous mutations are reported to be responsible for the antigenic change in FMDV [17,52]. The sporadic diversifying evolution is calculated using the dN/dS ratio, which has a risk of overestimation due to non-fixed mutations in the population. This bias is usually avoided by restricting the analysis to internal branches of the tree and by optimizing the branch lengths. However, despite the best efforts, there might be a chance of non-fixed mutations in the internal branches of a given dataset.

In our analysis, the transmission history of FMDV was generated using VP1 sequences only, as the number of sequences for other regions was limited to generate a comprehensive transmission history of different genomic regions. The phylogeographic analysis of the VP1 region inferred that the ancestor location of serotype A isolates might be Punjab, with a tree root in the year 2001 (72% posterior probability), while the ancestor location for SIS-13 isolates was also inferred as Punjab, with origin in 2012 (> 96% posterior probability) (Figure 3). The transmission network was reconstructed between seven discrete locations using the location-annotated maximum clade credibility (MCC) tree. The initial transmission route was between Punjab and Sindh, and the diffusion of viruses might has occurred between three locations: Punjab, Sindh, and ICT (95% credible set). The data showed that the virus intensity hub was Punjab, transmitting the virus to the areas of Sindh and ICT (Figure 6). The westward, northwest, and northeast transmission usually occurred from the eastern Punjab region.

The coalescent reconstruction of past demography for serotype A viruses revealed that N_e_, which is a function of genomic diversity [45], was fluctuating, and the highest peak was observed during 2012–2013; after 2015, a steady decline was observed until 2018, when the most recent samples of the current study were collected. The number of lineages was also increasing strongly until 2015. After that, it reached consistency. This may be due to the cyclic appearance of lineages and then their disappearance, leading to a temporal increase and decrease in genetic diversity [15]. Moreover, a decline in diversity could be due to disease control efforts concerted in time and space in the early years of the last decade. Several international and national collaborative projects were implemented in Pakistan to control FMD. In 2015, Pakistan advanced to stage two Food and Agriculture Organization of the United Nations (FAO) in the progressive control pathway for FMD. However, the vaccination of animals against FMD is still voluntary. Progressive control of FMDV program implementation in Pakistan [20] has some positive effects on disease control, but tackling virus spread and the resulting disease burden requires sustainable research efforts, which is rather challenging.

The dataset used in this study seems to suffer from study-centered sampling, with an under-representation of certain areas, such as GB, KPK, and Balochistan, in terms of sampling intensity, which can affect population-related interpretations. The understanding of virus sources and transmission between different regions can help in developing control strategies. However, current phylogenetic models can accommodate the uncertainty intrinsic in such datasets and can infer important transmission events [53,54]. Epidemiological information on serotype A, such as the number of outbreaks reported and species affected, is a limitation in the current study. Although this study has the constraint of reporting a limited number of complete genomes, it has the advantage of being a better tool to explore the evolutionary dynamics of endemic co-circulating strains. Moreover, sustained surveillance is important for determining vaccine efficacy and vaccine-induced shift in circulating viral strains.

## 5. Conclusions

The serotype A viruses in Pakistan have a strong temporal pattern in their genealogy. The sub-lineages are continuously evolving, with recombination in non-structural protein, and sporadic and pervasive diversifying selection acting on amino acid sites in structural proteins, contributing to genetic diversity. Serotype A is a rapidly evolving strain, and it has consequences for disease control in the region. Although VP1 sequences are commonly used for genotyping, complete genome sequences can better shed light on the evolution of co-circulating strains and their virulence. Continuous surveillance of FMDV in all provinces of Pakistan is essential to better understand the changing dynamics of this disease and virus evolution in this disease-endemic country.

## Figures and Tables

**Figure 1 viruses-14-01634-f001:**
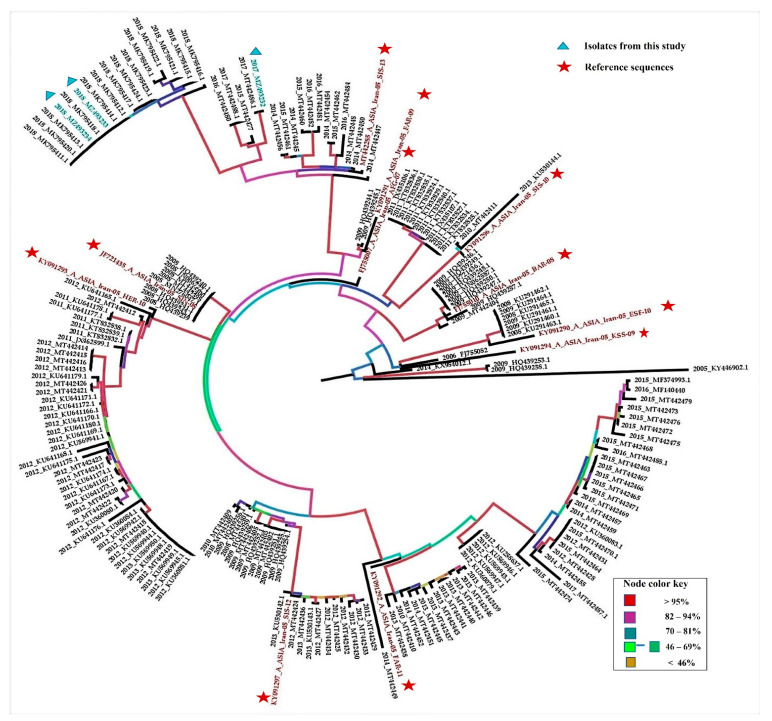
Mid-point rooted, maximum likelihood (ML) phylogenetic tree of serotype A isolates from Pakistan. The analysis includes 206 VP1 sequences and 11 reference sequences for sub-lineages of A/ASIA/Iran-05, shown in red color. FMDV isolates sequenced in this study are shown in green color. The node color represents the bootstrap support in percentages, color-coded according to the scheme given in the picture. Red color: >95%; purple color: 82–94%; blue color: 70–81%; green color (from dark to light shade): 69–46%; yellow color: <45%. Two sets of isolates did not cluster with closely related (>95% nucleotide similarity) reference sequence AFG-07; one set is supported by >95% bootstrap support, while the other has 70% support.

**Figure 2 viruses-14-01634-f002:**
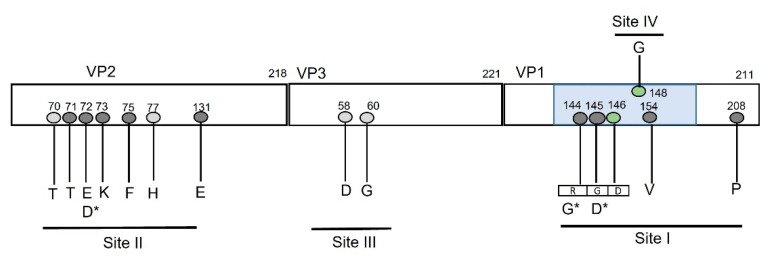
Comparison of antigenic sites of FMDV serotype A and their evolution under natural selection. The amino acid residues of each antigenic site are represented for each structural protein. The results of selection pressure estimation for each residue are mentioned. The light grey color indicated invariable residues; dark grey indicates sites under neutral evolution. The green color indicates residues under strong purifying selection. None of the residues were found under diversifying selection. * Substitution of two amino acids was observed at these positions: VP2 E72D is synonymous, while VP1 R144G and G145D are non-synonymous substitutions.

**Figure 3 viruses-14-01634-f003:**
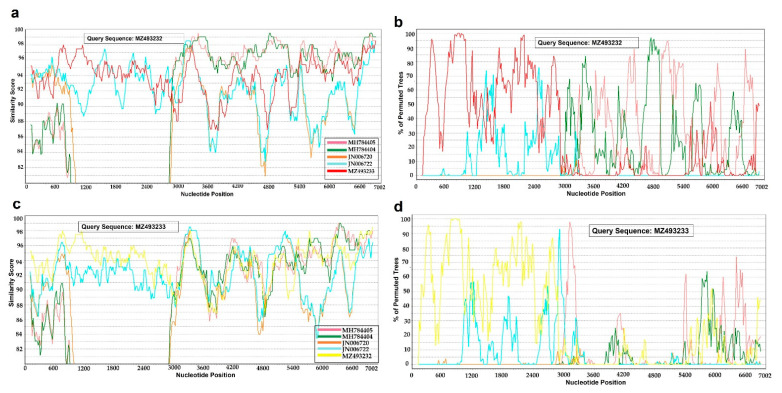
Recombination analysis of A/ASIA/Iran-05/SIS-13 isolates with most closely related isolates for serotype A, O, and Asia 1. Similarity plots were generated for query sequences (**a**) A_PK_C6_2017 and (**c**) A_PK_C8_2018. The nucleotide positions are drawn across the x-axis, while nucleotide similarity (percentage) is plotted along the y-axis. Breakpoints exist where sequences cross over each other. The figures show regions with more than 80% sequence similarity. The similarity was further tested by the permutation of trees with BootScan; plots (**b**,**d**) were generated with 1000 bootstrap for 100 permutation trees. The x-axis represents nucleotide positions, while the y-axis shows the percentage of permuted trees where selected sequences were closely related to the query. The threshold of the indication of a potential recombination event was set at 70%.

**Figure 4 viruses-14-01634-f004:**
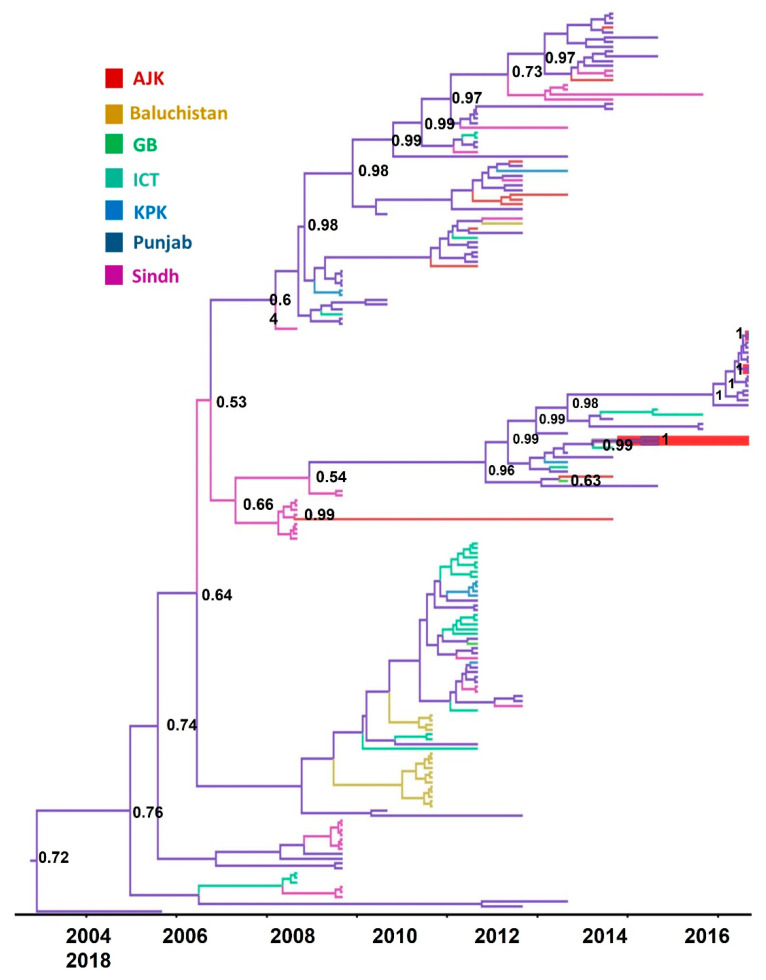
Reconstruction of phylogeographic diffusion of FMDV serotype A viruses from Pakistan. In total, 189 spatially and temporally stamped VP1 sequences were included in this analysis. The spatial dispersal of viruses in a discrete space was reconstructed to estimate the ancestral locations. The branches and nodes in the time-calibrated MCC tree are colored for locations and represent transmission history. The nodes are colored for ancestral locations and annotated with ancestral location probability. Taxa highlighted in red were sequenced in this study.

**Figure 5 viruses-14-01634-f005:**
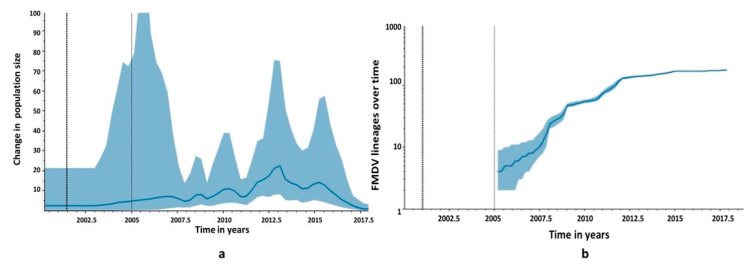
Reconstruction of the demographic history of FMDV serotype A in Pakistan. (**a**) Sky Grid Plot. The y-axis represents the change in effective population size (Ne) and the x-axis represents time in years. The peaks in population size are visible. The solid interval represents the 95% HPD support. (**b**) Lineages Through Time plot. The y-axis depicts the logarithmic increase in the number of FMDV lineages over time in years along the x-axis. The dark dotted line represents the mean tMRCA and the light dotted line represents 95% BCI.

**Figure 6 viruses-14-01634-f006:**
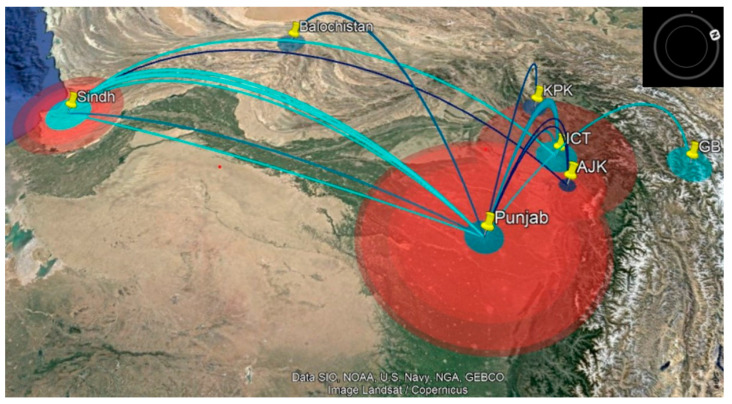
Visual reconstruction of spatiotemporal diffusion of FMDV serotype A viruses in Pakistan. The location-annotated MCC tree was used to depict the transmission of serotype A viruses between seven locations within the country. Each location is individually represented by a distinct color, and the route of transmission is represented by lines colored according to destination location or sink location. The polygons around each location represent the time spent by viruses in an area, along with the Bayesian confidence interval. Images rendered in Google pro-Earth; image: Landsat/Copernicus.

**Table 1 viruses-14-01634-t001:** A comparison of non-synonymous amino acid substitutions in the capsid proteins VP2 and VP3 of eight sub-lineages of A/ASIA/Iran-05.

Sub-Lineage	No. of Taxa	VP2 (AA = 218)									VP3 (AA = 221)											
Amino Acid Site No.		38	64	79	98	86	133	134	195	207	3	59	65	70	92	94	95	99	117	159	175	220
AFG-07	1 Taxon	T	K	E	F	E	S	P	S	F	V	D	V	E	S	I	A	T	S	S	V	A
HER-10	16 Taxa		Q					S	P		L	N	E									
BAR-08	2 Taxa			V	Y									D							A	
SIS-13	MZ493232					D		S		S							S		A			T
SIS-13	MZ493233, MZ493234												M					A		P		
FAR-11	1 Taxon						A		T						A							
FAR-09	1 Taxon							L														
ESF-10	6 Taxa	I												D		L						T
SIN-08	4 Taxa													D								T

Singleton non-synonymous substitutions found in only one isolate of a set of sequences are not listed. Empty cells have the same substitutions as those listed in the top cell for AFG-07. Isolates belonging to SIS-13 were sequenced in this study and were submitted to GenBank under Accession numbers MZ493232, MZ493233, and MZ493234.

**Table 2 viruses-14-01634-t002:** A comparison of non-synonymous amino acid substitutions in capsid protein VP1 of eight sub-lineages of A/ASIA/Iran-05.

Sub-Lineage	No. of Taxa	VP1 (AA = 211)																										
Amino Acid Site No.		17	24	28	43	45	46	65	83	95	96	99	109	134	135	140	141	142	145	149	154	155	160	168	171	196	198	204
AFG-07	1 Taxon	N	A	H	N	V	S	L	D	V	G	A	K	S	K	S	G	G	G	S	V	A	S	R	T	S	Q	Q
HER-10	16 Taxa																						G	K				
BAR-08	2 Taxa	D	V			A		F	E	E	M	N				G												R
SIS-13	MZ493232						G							N		G		S		P	I							R
SIS-13	MZ493233 MZ493234			Q						E	E		R															
FAR-11	1 Taxon		T			A				E			R			G									I	L	S	
FAR-09	1 Taxon		V								E				R	G		S	D	L		T						R
ESF-10	6 Taxa		T	Q	S				E		E				R		S		D							A		
SIN-08	4 Taxa														R	G								Q				

Singleton non-synonymous substitutions found in only one isolate of a set of sequences are not listed. Empty cells have the same substitutions as those listed in the top cell for AFG-07. Isolates belonging to SIS-13 were sequenced in this study and were submitted to GenBank under Accession numbers MZ493232, MZ493233, and MZ493234.

## Data Availability

The sequences generated in this study are deposited in NCBI GenBank.

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
