# Peer review of "Evolutionary Dynamics of Foot and Mouth Disease Virus Serotype A and Its Endemic Sub-Lineage A/ASIA/Iran-05/SIS-13 in Pakistan"

_viruses, 2022, doi:10.3390/v14081634_

Round 1

Reviewer 1 Report

In this manuscript, Naqvi et al. report 3 near full-length genome sequences of two FMDV isolates from Pakistan. Together with publicly available sequences, the authors reconstructed the genomic epidemiology of FMDV serotype A in Pakistan. I can see that the authors have addressed some of my concerns in the cover letter. However, even after the revision, the analyses and discussion still seem to be more or less superficial and inappropriate. The results and the quality of the paper might not represent the advances and the standard required for a Viruses paper.

Comments:

In general, the manuscript is prepared rather poorly. Almost all of the figures presented are of poor qualities and poorly formatted, and several figure legends lack sufficient details. To give a concrete example, besides its poor quality, Figure 1 does not even indicate where the 3 sequenced genomes are in the tree. Sequences’ names on the tree are not at all legible, and its legend is lacking details. Table S2 (referred to on the line 278) and File S1 (referred to on the line 280) are missing from the supplementary file. Moreover, the manuscript still contains numerous grammatical errors and imprecise sentences (one of my previous comments that are not addressed by the authors in the cover letter). Here are some examples from the first paragraph of the main text:

Line 34: “a positive sense single stranded” -> “a positive-sense single-stranded”

Lines 35-36: “acute vesicular disease” -> “an acute vesicular disease”

Line 36: “FMDV has diverse host range” -> “FMDV has a diverse host range”

Lines 37-38: “transmission” -> “transmissions”

Line 38: “Middle East” -> “the Middle East”

Line 39: “world reference laboratory for FMD” -> “the World Reference Laboratory for FMD”

Line 40: “endemic pool 3” -> “the endemic pool 3”

Lines 40-41: “three serotypes of FMDV O, A, and Asia 1” -> “three serotypes of FMDVs, including O, A, and Asia 1”

Line 42: “in various studies [3].” -> although the authors used the phrase “various studies”, only one study is cited.

Some technical terms are also used incorrectly. For example, the term “sequence homology” is used inappropriately to mean “sequence similarity” in the discussion (lines 390-402). There are many more errors throughout the manuscript, which I won’t go into details here.

Regarding the genomic epidemiological analysis, while the recombination analysis showed that some genome regions of A_PK_C6_2017 are more similar to Serotype O than Serotype A viruses (lines 282-286), their analyses mostly considered only Serotype A viruses, and thus the results are fragmented. Furthermore, in the discussion, while the results are sufficiently described, the discussion is quite superficial. For example, the authors described that “[r]econstruction of past demography for serotype A revealed that Ne… was fluctuating till 2015 when a steady decline was observed which stopped in 2018…., a decline in diversity could be due to disease control efforts concerted in time and space.” – here, the authors should find out what actually happened in 2015 to explain the drop in the diversity, rather than just proposing a hypothesis. Furthermore, I do not quite understand why the authors mentioned that the decline stopped in 2018. Base on the tree (Figure 4) and the Ne graph (Figure 5a), it can be inferred that the trend stopped simply due to the lack of data rather than anything else; and thus, the text is misleading and confusing.

Regarding the natural selection analysis, I recommended that some other kinds of analyses, rather than analysis of dN/dS ratios, are needed to quantify evolutionary pressure in a virus population, and suggested the authors to see the paper “the Population Genetics of dN/dS” by Sergey Kryazhimskiy and Joshua B. Plotkin for a discussion why dN/dS would be a poor indicator of selection pressure at the population level. The authors replied that “we read the suggested article by the esteemed reviewer, but this article assumes for example free combination etc.” Indeed, the article does not consider recombination in the discussion, but that was not the point. I did not recommend the paper for the authors to look for a way for estimating evolutionary pressure that can accommodate recombination, but just to point out that dN/dS would not be good in this context. In fact, recombination would make interpretation of dN/dS ratios estimated from a population dataset even more difficult.

As an attempt to address my concern, the authors restricted the calculation of dN/dS ratios “to internal branches to avoid dN/dS ratio bias on population evolution inference due to transient changes in external branches of phylogeny that are not yet fixed by selection [30].” (lines 199-201). Firstly, the paper [30] “Persistent HIV-1 replication maintains the tissue reservoir during therapy” is not appropriate as a reference for this sentence. Secondly, while it may be true that excluding terminal branches from the calculation of dN/dS ratios can reduce the effects of non-fixed mutations to some degrees, it still by no means justifies the use of dN/dS ratios in this context. Considering the dataset used in this study, it is still very likely that (relatively shallow) internal branches still contain a high number of non-fixed mutations, and the problem likely still persists.

The authors “also used polymorphism frequency spectrum analyzing tests, Tajima’s D, Fu and Li's D and F methods in DnaSP V 6 [to] make inferences about deviation from neutrality based on low, intermediate and high frequency polymorphism” (lines 202-204). The authors found that “[t]he results of Tajima’s D, Fu and Li's D and F methods were non-significant with p value > 0.2.” These results implied that overall there is no evidence for selection, indeed conflicting with the results from dN/dS analysis, but the authors instead ignored the results and “did not consider them for selection inference.” (lines 323-325). This is highly inappropriate and scientifically unjustified.

Overall, the results and the quality of the paper might not represent the advances and the standard required for a Viruses paper. 

Author Response

Please see the attached file for point by point reply to the reviewer's suggestions and comments. English language and style have been thoroughly revised. 

Reviewer 2 Report

The questions are well-resolved and may be published in their current form.

Author Response

There were no specific points raised by reviewer 2 therefore no response file is submitted

Reviewer 3 Report

Spontaneous mutations are a common characteristic of the foot and mouth disease virus (FMDV). So https://doi.org/10.3390/ani11061697  this recent study will be useful in your manuscript

All Figs.  need more resolution 

Add conclusion sevtion

Author Response

Please, find attached the file with point by point reply to the reviewer 3 comments.

Round 2

Reviewer 1 Report

The authors have addressed all of my concerns.

This manuscript is a resubmission of an earlier submission. The following is a list of the peer review reports and author responses from that submission.

Round 1

Reviewer 1 Report

The complete genomic sequences of three FMDV serotype A sub-lineage SIS-13 circulating in Pakistan were determined by the authors.

However, the majority of the analysis was focused on the VP1 region.

Full genome sequences, in whatever numbers are available, can be examined to identify if any significant changes in antigenic and functional motifs have occurred.

At the very least, I recommend comparing the P1 region.

Organize the results section according to the material and method used.

A few small changes

Line 34: replace the word ' Picornavidae' with 'Picornaviridae'.

Line 35: FMDV is not simply a problem for ruminants; alternatively, utilize cloven-hoofed animals.

Lines 185-187 is unclear. Please rephrase the sentence.

Reviewer 2 Report

The study describes lineage reconstruction of a circulating strain of FMDV. Standard methods were used for virus culture and sequencing. Overall, it is a nicely conducted study and it can be accepted in its present form.

Reviewer 3 Report

Naqvi et al. report 3 near full-length FMDV genome sequences from Pakistan, all of which belong to serotype A. Together with publicly available sequences, the authors reconstruct the genomic epidemiology of FMDV serotype A in Pakistan.

Overall, the manuscript is rather poorly written, containing numerous grammatical errors and imprecise sentences, but perhaps most worrying is the research methodology employed by the authors. It is well known that recombination is a very common feature of FMDV, and indeed the authors did detect it in their dataset. It is also well known that, in phylogenetic reconstruction, the presence of recombinants / recombination regions in the dataset can severely distort the estimated tree topology and branch lengths. While the authors did screen for potential recombinants in their dataset and did identify recombination regions (lines 156-160; 207-225), based on the description of the methods and the results in the manuscript, it is unclear if the authors did address this issue appropriately or not. Even if the authors did use recombination free sequences to reconstruct their tree, the fact that there is just only one tree in the paper indicates that the authors simply assume that the history of that one genomic region is the history of the virus transmission. What about the histories of other genomic regions? To reconstruct the genomic epidemiological history of a virus with a high recombination rate like FMDV, histories of multiple genomic regions should be considered. Furthermore, there are several issues with their phylogenetic analysis. For example:

Line 132: “Problematic sequences were removed to improve the root-to-tip correlation coefficient.” => At face value, this is highly inappropriate and unscientific. The details of the problems are missing and the removal of the sequences does not seem justified.

Line 136: “The SRD06 substitution model, uncorrelated relaxed clock lognormal molecular clock13 and nonparametric coalescent Bayesian skygrid demographic model23 (42 transition points) were used and SIS-13 taxa were specified with a monophyletic constraint enabled.” => The justification for these settings is missing, in particular the monophyletic constraint.

Line 143: “The Four independent chains of Markov chain Monte Carlo (MCMC) were run for 50 million iterations and data logs were combined in Log Combiner v 1.10.4 to achieve effective sample size (>200) for desired parameters in trace file” => This is inappropriate. As nearly all parameters interact with one another in some ways, posterior distributions of almost all parameters, and not just those that are interested by the authors, should be well sampled and well mixed to be sure that the answers are justified.

Moreover, the authors quantify natural selection pressure by using the dN/dS ratio (lines 160-165). However, it is well known that the dN/dS ratio is not reliable for quantifying selection pressure at the population-genetics level. Interpretation of the dN/dS value as an indicator of the direction and strength of natural selection is only valid when almost all genetic variations in the sequences examined are fixed substitutions. See “The Population Genetics of dN/dS” by Sergey Kryazhimskiy and Joshua B. Plotki, for example. Other kinds of analyses are needed to quantify evolutionary pressure in this context.

The method employed to examine the genetic diversity of the virus is also very naïve and primitive (lines 166-174; 177-191; 215-223), in particular when they have a phylogenetic tree in their result. Although not necessarily incorrect, the results regarding this part are also described rather confusingly, low-level, and are difficult to assess with minimum discussion.